# Targeting HBV cccDNA Levels: Key to Achieving Complete Cure of Chronic Hepatitis B

**DOI:** 10.3390/pathogens13121100

**Published:** 2024-12-13

**Authors:** Wei He, Zhijin Zheng, Qian Zhao, Renxia Zhang, Hui Zheng

**Affiliations:** 1Jiangsu Key Laboratory of Infection and Immunity, Institutes of Biology and Medical Sciences, Soochow University, Suzhou 215123, Jiangsu, China; hw18862232157@163.com (W.H.); 20234052010@stu.suda.edu.cn (Z.Z.);; 2MOE Key Laboratory of Geriatric Disease and Immunology of Ministry of Education of China, Collaborative Innovation Center of Hematology, International Institute of Infection and Immunity, Institutes of Biology and Medical Sciences (IBMS), School of Medicine, Soochow University, Suzhou 215123, Jiangsu, China; 3Department of Laboratory Medicine, Institute of Laboratory Medicine, Sichuan Provincial People’s Hospital, School of Medicine, University of Electronic Science and Technology of China, Chengdu 611731, Sichuan, China

**Keywords:** HBV, chronic hepatitis B, cccDNA, CRISP-cas9, interferon

## Abstract

Chronic hepatitis B (CHB) caused by HBV infection has brought suffering to numerous people. Due to the stable existence of HBV cccDNA, the original template for HBV replication, chronic hepatitis B (CHB) is difficult to cure completely. Despite current antiviral strategies being able to effectively limit the progression of CHB, complete CHB cure requires directly targeting HBV cccDNA. In this review, we discuss strategies that may achieve a complete cure of CHB, including inhibition of cccDNA de novo synthesis, targeting cccDNA degradation through host factors and small molecules, CRISP-Cas9-based cccDNA editing, and silencing cccDNA epigenetically.

## 1. Introduction

Chronic hepatitis B (CHB) is a chronic inflammation of the liver induced by hepatitis B virus (HBV) infection. Over 250 million people globally are living with CHB, which is responsible for 83% of deaths from viral hepatitis [1,2]. The vast majority of patients require lifelong therapy. Despite the availability of effective HBV vaccines for several decades, CHB continues to be a significant public health burden worldwide. As CHB progresses, patients are at an increased risk of evolving further hepatic diseases, such as hepatocellular carcinoma. Annually, about 1 million people die from CHB and its complications [3]. Current treatments approved by the U.S. Food and Drug Administration, such as pegylated interferon-alpha (PEG-IFNα) and nucleos(t)ide analogues (NAs), fail to completely cure CHB [4,5,6]. Consequently, researchers continue to focus on discovering new therapeutic options for CHB.

The HBV genome consists of relaxed-circular DNA (rcDNA), a form of partially double-stranded DNA [7]. During viral infection, HBV initially binds to the low-affinity receptor heparan sulfate proteoglycan receptor (HSPG) on the hepatocyte membrane. This initial binding is followed by the interaction of the HBV Pre-S1 region with the sodium taurocholic co-transporting polypeptide (NTCP) [8,9]. Subsequently, HBV enters hepatocytes via endocytosis facilitated by epidermal growth factor receptor (EGFR) [10]. The envelope is discarded inside the cytoplasm, and the nucleocapsid is transported to the nuclear pore later where it disassembles. The rcDNA then enters the nucleus, where it is converted into fully double-stranded DNA with the help of host proteins, forming covalently closed circular DNA (cccDNA). cccDNA utilizes host histones to assemble into a minichromosome, becoming a stable part of the nucleus [7]. The stability of cccDNA poses significant challenges for the immune system to effectively target and eliminate it, complicating the development of effective therapies against cccDNA [11,12,13].

cccDNA has four open reading frames, including the S, C, P, and X regions, which generate four types of mRNAs (3.5 kb, 2.4 kb, 2.1 kb, 0.7 kb). The 3.5 kb pgRNA is the template for reverse transcription into rcDNA [14]. These transcripts are responsible for producing seven viral proteins: secreted antigen HBeAg, nucleocapsid protein HBcAg, surface antigens Pre-S1, Pre-S2, HBsAg, transcriptional activator HBx, and DNA polymerase pol/RT [15,16,17,18]. Nucleos(t)ide analogues (tenofovir, entecavir) can restrict the reverse transcription of pgRNA and inhibit HBV transcription, but they cannot prevent the conversion of rcDNA into cccDNA or affect the cccDNA already stabilized in the nucleus. CHB patients may experience a relapse after treatment interruption, and most of them require long-term treatment. The role of interferon in inhibiting HBV remains not yet fully understood. Interferon-stimulated genes (ISGs) might play a role in suppressing HBV transcription or in directly targeting cccDNA. However, the poor tolerance and potential side effects of interferon limit its application in long-term treatment [2,19,20,21].

The cure for chronic hepatitis B is divided into functional cure and complete cure. A functional cure is evidenced by the absence of HBsAg and HBV DNA in the serum after treatment, coupled with a significantly reduced risk of hepatocellular carcinoma over time [22]. In contrast, a complete cure is defined as the total eradication of HBsAg, cccDNA, and integrated HBV DNA [23]. At present, only a functional cure of HBV can be reached clinically. Indirect therapeutic strategies, such as HBV entry inhibitors, core synthesis inhibitors, and capsid assembly modulators, are difficult to achieve a complete cure for CHB. The inherent stability of cccDNA within hepatocytes poses a major challenge to this.

cccDNA represents a central component of the HBV lifecycle, serving as the primary transcription template for all HBV transcripts, such as pre-genomic RNA (pgRNA). This template is crucial for the production of HBV proteins and the generation of new virions. The persistence and stability of cccDNA within hepatocytes make it the greatest obstacle to achieving a complete cure for CHB. Owing to the critical role of cccDNA in HBV transcription and replication, it is essential to identify molecules or drugs that can directly or indirectly degrade cccDNA or permanently silence it. This review discusses the latest advancements in strategies aimed at combating cccDNA, including synthesis inhibitors of cccDNA, cccDNA eliminators, cccDNA gene editing strategies, intracellular mechanisms of cccDNA degradation, and cccDNA transcriptional silencing, in order to explore the feasible development direction for the complete cure of chronic hepatitis B.

## 2. Mechanisms and Inhibitors of cccDNA Synthesis

The conversion of rcDNA into cccDNA is a complex process that is not fully understood. In the nucleus, the HBV DNA polymerase attached to the 5′ end of the antisense strand is first removed, resulting in deproteinized rcDNA (DP-rcDNA), a step in which tyrosyl-DNA phosphodiesterase 2 (TDP2) is involved (Figure 1) [24]. However, the function of TDP2 in this deproteinization procedure remains debated, as some studies suggest it may not be involved in rcDNA repair mechanisms [25,26]. Following deproteinization, the DNA flap and RNA primers are removed, and host DNA polymerases repair the rcDNA gap to form cccDNA. In this process, the DNA polymerases Pol α and Pol κ may be necessary host factors [27,28]. The final step, which involves sealing the rcDNA gap, likely requires DNA ligases 1 and 3 (LIG1/3). Inhibitors of these ligases, such as L1, L25, and the pan-DNA ligase inhibitor L189, have been shown to limit cccDNA formation [29]. Flap endonuclease 1 (FEN-1) plays a key role in the disengagement of the 5′ DNA flap, facilitating the conversion of rcDNA to cccDNA. The FEN-1 inhibitor 3-Hydroxy-5-methyl-1-phenylthieno [2,3-d] pyrimidine-2,4 (1H,3H)-dione (PTPD) has been found to reduce cccDNA levels [30]. Furthermore, DNA repair factor poly(ADP-ribose) polymerase 1 (PARP1) participates in cccDNA generation, with the PARP inhibitor olaparib having been shown to inhibit cccDNA production, highlighting the importance of poly(ADP-ribosylation) in this process [31].

Five key components involved in DNA lagging strand synthesis are essential for cccDNA formation, representing the minimal necessary factors for this synthesis [32]. Moreover, the DNA polymerase inhibitor aphidicolin has been found to reduce cccDNA levels in HBV-infected cells [32]. Interestingly, studies suggest that the repair mechanisms for the sense and antisense strands of rcDNA are different. While the repair of the sense strand requires five lagging strand synthesis factors, including proliferating cell nuclear antigen (PCNA), the repair of the antisense strand depends only on FEN-1 and LIG1 [33].

At present, some cccDNA synthesis inhibitors have been reported, but the precise mechanisms of these inhibitors warrant further study. Two disubstituted sulfonamides (DSS), CCC-0975 and CCC-0346, were determined to be cccDNA synthesis inhibitors (Figure 2). These compounds reduce the levels of both cccDNA and DP-rcDNA without affecting their elimination kinetics, suggesting they may inhibit cccDNA formation by disrupting the production of DP-rcDNA [34]. However, the specific mechanism by which they affect DP-rcDNA remains unclear. A study revealed the presence of a casein kinase II (CKII) phosphorylation site in the HBV polymerase. Inhibition of CKII using 2-dimethylamino-4,5,6,7-tetrabromo-benzimidazole (DMAT) prevented the nuclear import of the polymerase-rcDNA complex, thereby inhibiting cccDNA generation [35]. Additionally, the ataxia-telangiectasia mutated proteins (ATM) and Rad3-related (ATR) pathway are involved in the repair of the rcDNA antisense strand. Inhibitors of the ATR-CHK1 pathway have been shown to significantly inhibit cccDNA formation. Furthermore, inhibition of this pathway results in the accumulation of a novel rcDNA intermediate with a 5′ truncated antisense strand, indicating that the ATR pathway is integral to the conversion of rcDNA into cccDNA (Figure 2) [36]. Although these cccDNA synthesis inhibitors can effectively block the conversion of rcDNA to cccDNA and prevent replenishment of the cccDNA pool, they do not affect existing cccDNA within the nucleus. Therefore, further research is needed to develop strategies that directly target and eliminate established cccDNA.

## 3. Targeting cccDNA Degradation

### 3.1. Host Factors Target cccDNA for Degradation

Apolipoprotein B mRNA-editing enzyme 3 (APOBEC3) family members play a key role in recognizing foreign DNA and mediating its deamination, which leads to C-to-T or G-to-A hypermutations, thereby protecting the host from various viral infections, including HIV-1, HTLV-1, and HPV [37]. For instance, APOBEC3A is induced by interferon in response to foreign DNA detection and deaminates cytidine in double-stranded DNA into uridine, an atypical nucleoside. The uridine is then processed by uracil DNA glycosylase (UNG2) into an apurinic/apyrimidinic site (AP site), ultimately resulting in the degradation of the foreign DNA. Endonuclease apurinic/apyrimidinic endonuclease (APEX) may serve as the final effector in this process, though other intracellular nucleases, such as three prime repair exonuclease 1 (TREX1) and DNase1/2, may also contribute [38]. In a study by Lucifora et al. [39], IFN-α and lymphotoxin-β receptor activation separately induced the production of APOBEC3A or APOBEC3B, which resulted in the deamination of cccDNA and the formation of AP sites, leading to cccDNA degradation. Importantly, this effect did not induce hepatocellular toxicity or affect genomic DNA. Subsequently, research by Xia et al. [40] demonstrated that IFN-γ and TNF-α released by T cells could degrade cccDNA without causing hepatocyte death, an effect also linked to the deamination activity of APOBEC3A/B.

The precise mechanism by which cccDNA is degraded after deamination and AP sites formation remains unclear. In the experiments conducted by Xia et al. [40], knockdown of APEX1 restored cccDNA levels following treatment with IFN-γ and TNF-α, indicating that APEX1 promotes cccDNA degradation. However, APEX1 knockdown did not completely reverse cccDNA downregulation, suggesting that other nucleases may also contribute to its degradation. Subsequently, Stadler et al. [41] identified ISG20 as the only interferon-stimulated molecule with nuclease activity that is upregulated by IFN-α and IFN-γ. Knockdown of ISG20 significantly inhibited IFN-induced degradation of cccDNA, while co-expression of APOBEC3A with ISG20 significantly degraded cccDNA (Figure 2). Notably, interferon treatment or co-expression of APOBEC3A and ISG20 had no effect on the cccDNA of HBX-deficient HBV. This suggests that cccDNA degradation is restricted to cccDNA with transcriptional activity, likely because APOBEC3A primarily targets single-stranded DNA. The partial single-stranded region that arises during cccDNA transcription provides the opportunity for APOBEC3A to recognize and deaminate cccDNA.

### 3.2. Gene Editing for cccDNA

Since the introduction of CRISPR-Cas9 gene editing, this technology has been explored across various clinical fields [42]. By using a guide RNA (gRNA) to target specific DNA sequences, the CRISPR-Cas9 system directs the Cas9 protein to create double-strand breaks (DSBs) at the target site. Cells typically repair these breaks through non-homologous end joining (NHEJ), which disrupts the target sequence and leads to its destruction [43,44]. Studies have shown that delivering the CRISPR-Cas9 expression vector via hydrodynamic injection can significantly enhance the clearance of cccDNA [45,46,47]. However, several critical challenges remain before CRISPR-Cas9 can be effectively applied in clinical settings.

One of the challenges with CRISPR-Cas9 is the potential for off-target effects. It is crucial to ensure that the guide RNA (gRNA) designed for cccDNA targets only the intended sequence and avoids unintended binding to the host’s genomic DNA. Meanwhile, HBV DNA is possibly integrated into the host genome, and using CRISPR-Cas9 to target cccDNA carries the risk of inducing chromosomal translocations. However, emerging gene-editing techniques offer promising solutions to these concerns. Yang et al. [48] developed a CRISPR/Cas9-mediated non-cleaving base editor, which introduced nonsense mutations into both integrated HBV DNA and cccDNA, significantly reducing HBsAg secretion and HBV replication. Similarly, Smekalova et al. [49] employed a cytosine base editor (CBE) to induce C-to-T mutations in both cccDNA and integrated HBV DNA, effectively halting HBV replication and silencing viral protein expression (Figure 2). These base-editing approaches inactivate cccDNA without creating DSBs in HBV-infected hepatocytes, offering a safer and more precise alternative to traditional CRISPR-Cas9 methods.

Achieving lasting therapeutic effects while minimizing off-target risks requires highly efficient and precise CRISPR-Cas9 delivery systems that can specifically target infected liver cells. This is a significant obstacle to the use of CRISPR-Cas9 in CHB therapy. Zhang et al. [50] synthesized a guide RNA (gRNA)/Cas9 ribonucleoprotein (RNP) as a nonviral agent, developing a novel CRISPR/ Cas9-mediated gene therapy. Similarly, the lipid nanoparticles (LLNs) developed by Jiang et al. [51] can effectively deliver Cas9 mRNA and single-guide RNA (sgRNA) to the liver, resulting in successful editing of HBV DNA, reductions in HBsAg, HBeAg, and other HBV markers, all while maintaining a low off-target rate. In another advancement, Wang et al. [52] developed NIR-responsive biomimetic nanoparticles (UCNPs-Cas9@CM), which efficiently delivered Cas9 RNP into liver cells, leading to effective HBV DNA editing. Through this system, HBsAg, HBeAg, HBV pgRNA, HBV DNA, and cccDNA levels were significantly reduced. Additionally, Zeng et al. [53] engineered extracellular vesicles containing Cas9/gRNA complexes along with vesicular stomatitis virus glycoprotein (VSV-G), which showed efficacy in reducing viral antigens and cccDNA levels. These innovative delivery systems represent promising steps toward more precise and effective CRISPR-Cas9-based therapies for HBV.

### 3.3. cccDNA Reducers

Several small-molecule compounds have shown potential in eliminating cccDNA, though their precise mechanisms of action remain largely unknown. Chen et al. [54], for example, conducted high-throughput screening using primary human hepatocytes (PHH) and identified xanthone derivatives as novel cccDNA eliminators. Among these, compound 59 demonstrated significant potency, good oral bioavailability, and low cytotoxicity, effectively reducing preexisting cccDNA in HBV-infected cells. Similarly, Wang et al. [55] screened 84,600 compounds and discovered that ccc_R08 acts as a novel reducer of cccDNA. PHH treated with ccc_R08 two days after HBV infection exhibited dose-dependent reductions in HBV DNA, HBsAg, and HBeAg levels. More importantly, ccc_R08 selectively reduced cccDNA levels without evident cytotoxicity. In HBVcircle model mice, oral administration of ccc_R08 twice daily significantly decreased HBV DNA and antigen levels, showcasing its anti-HBV efficacy. However, further studies are required to assess its safety and clarify whether ccc_R08 directly targets cccDNA, as its mechanism remains uncertain. Another promising compound, PAC5 (a natural sesquiterpenoid phyllanthacidoid A derivative), a heterogeneous nuclear ribonucleoprotein A2B1 (hnRNPA2B1) agonist, was shown to eliminate HBV cccDNA and decrease large antigen expression. Mechanistically, PAC5 binds to hnRNPA2B1, facilitating its translocation to the cytoplasm and activating the TBK1-IRF3 pathway, resulting in the production of type I interferon [56]. Thus, PAC5’s anti-HBV effect is dependent on the IFN signaling pathway. Moreover, Ren et al. [57] developed a cccDNA-targeted screening platform using HBV acceptor cells with high cccDNA loads, enabling continuous HBV production. Through this platform, they identified several antihistamines, H1 antagonist Bilastine, H2 antagonist Nizatidine, and H3 antagonist Pitolisant, respectively, which downregulated cccDNA levels. The inhibition of cccDNA by these antihistamines is unlikely to be mediated directly by histamine receptor signaling. Instead, the authors hypothesize that the reduction of cccDNA may be due to a variety of mechanisms, such as inhibiting HBV entry or interfering with cccDNA synthesis.

## 4. Targeting cccDNA for Transcriptional Silencing

### 4.1. cccDNA Epigenetic Silencing

cccDNA forms a highly stable minichromosome by integrating host histone and non-histone proteins, making it susceptible to epigenetic regulation. This provides an opportunity to silence its transcription by modifying the associated histones (Figure 2). Acetylation of histone 3 (H3) and trimethylation of H3K4 (H3K4me3) are considered markers of transcriptional activation for cccDNA [58,59]. Histone deacetylase 11 (HDAC11) suppresses cccDNA transcription by reducing the acetylation of H3K9 (H3K9ac) and H3K27 (H3K27ac) [59]. Interestingly, miRNA-548ah inhibits HDAC4, which leads to increased cccDNA replication, suggesting that HDAC4 may regulate the deacetylation of histones associated with cccDNA [60]. On the other hand, the trimethylation of H3K9 (H3K9me3), mediated by SET domain bifurcated histone lysine methyltransferase 1 (SETDB1), signals transcriptional repression of cccDNA and induces its epigenetic silencing [61]. Similarly, the symmetric dimethylation of arginine 3 on histone H4 (H4R3), a mark of transcriptional inhibition, is regulated by Protein Arginine Methyltransferase 5 (PRMT5) [62]. WD Repeat Domain 77 protein (WDR77) enhances PRMT5-mediated symmetric dimethylation of H4R3 on cccDNA minichromosomes, further restricting its transcription. The viral protein HBx counteracts these repressive mechanisms by using a damage-specific DNA binding protein 1 (DDB1)-containing E3 ubiquitin ligase to degrade WDR77, impairing PRMT5’s methyltransferase function, and promoting cccDNA transcription [63]. In a related mechanism, PRMT1, another methyltransferase, also suppresses HBV transcription. However, HBx can bind to PRMT1, downregulating its activity and further enhancing viral transcription [64].

The sirtuin (SIRT) family, a group of NAD^+^-dependent histone deacetylases, plays diverse roles in regulating cccDNA transcription. Members of the family influence HBV transcription and replication in distinct ways. SIRT1 is crucial for maintaining cccDNA transcriptional activity. Its absence significantly reduces downstream cccDNA transcripts and viral protein levels. SIRT1 enhances the activity of the HBV core promoter through its interaction with the transcription factor AP-1, and inhibition of SIRT1 with sirtinol leads to a marked reduction in cccDNA transcription [65]. SIRT2 promotes HBV transcription and replication by heightening the activity of HBV enhancers and promoters through targeting transcription factor p53 [66]. Silencing SIRT2 suppresses HBV transcription and replication, and the allosteric inhibitor FLS-359 inhibits the initial synthesis of cccDNA while impairing its transcriptional activity [67]. In contrast, SIRT3 acts as a limiting factor for cccDNA transcription. SIRT3 is recruited to deacetylate cccDNA H3K9, facilitating the recruitment of histone methyltransferase SUV39H1 and decreasing the association of SET domain containing 1A (SETD1A) with cccDNA. This results in increased H3K9me3 (a repressive mark) and reduced H3K4me3 (an activating mark) on cccDNA, effectively silencing its transcription [68]. In summary, while SIRT1 and SIRT2 promote cccDNA transcription and HBV replication, SIRT3 functions as a suppressor, indicating that sirtuins play both activating and inhibitory roles in the regulation of cccDNA.

### 4.2. Transcription Factors Regulating cccDNA

Hepatitis B virus (HBV) relies on host transcription factors to facilitate cccDNA transcription and viral replication, making these factors potential therapeutic targets. One major transcription factor is hepatocyte nuclear factor 4α (HNF4α), which is important for HBV transcription by binding to the precore/core promoter and activating the synthesis of pgRNA [69]. Spliceosome-associated factor 1 (SART1) downregulates HNF4α by binding to the P1 promoter of HNF4α, thereby restricting cccDNA transcription [70]. Additionally, the interaction between estrogen receptor α (ER-α) and HNF4α prevents HNF4α from binding to HBV enhancer I, reducing HBV transcription. This may explain why women with HBV infection tend to have lower viral loads and reduced rates of liver cancer compared to men [71]. High mobility group AT-hook 1 (HMGA1) also supports HBV transcription and replication. HMGA1 recruits forkhead box O3alpha (FOXO3α) and peroxisome proliferator-activated receptor-gamma coactivator-1 alpha (PGC1α) to enhance viral transcription. Meanwhile, HBx interacts with the transcription factor SP1 to increase HMGA1 expression, creating a positive feedback loop that further promotes viral replication [72].

cccDNA typically exists as a stable episomal minichromosome within the nuclei of infected hepatocytes. However, research has shown that Yin Yang 1 (YY1), along with HBx, mediates the recruitment of cccDNA minichromosomes to the 19p13.11 region of the human chromosome, contributing to cccDNA transcriptional activation [73]. Another key factor, E2F4, directly binds to cccDNA and activates HBV core promoters. The highly conserved TTAAAGGTC sequence in the HBV genome serves as E2F4’s binding site. However, E2F4’s ability to promote HBV transcription is moderated by the Inhibitor of Differentiation-1 (Id1), which forms heterodimers with E2F4, preventing its function and thereby inhibiting viral transcription and replication [74].

### 4.3. Targeting HBx

The HBx protein, a non-structural regulatory protein of HBV, plays a crucial role in HBV replication by binding to cccDNA minichromosomes. HBx is instrumental in initiating and maintaining cccDNA transcription, while also employing various mechanisms to counteract host factors that restrict HBV replication and upregulate host proteins that favor viral replication. One such host defense mechanism involves the Structural Maintenance of Chromosomes (SMC) protein complex, particularly SMC5/6, which binds to episomal cccDNA minichromosomes and inhibits their transcription [75]. To overcome this, HBx recruits DNA damage-binding protein 1 (DDB1), forming the HBx-DDB1-CUL4-ROC1 (CRL4) E3 ligase complex, which targets SMC5/6 for degradation [75,76,77]. By degrading SMC5/6, HBx removes this inhibitory factor and facilitates HBV gene expression. Treatment with siRNA targeting all HBV transcripts or PEG-IFNα significantly reduces HBx levels, allowing SMC5/6 to reappear in liver cells and resume its interaction with cccDNA, thus inhibiting viral transcription once again (Figure 2) [78].

Another mechanism of HBx regulation involves the E3 ubiquitin ligase TRIM21, which interacts with HBx, promoting its ubiquitination and proteasomal degradation. This action prevents HBx-induced degradation of SMC5/6, thereby limiting HBV replication [79]. Mitochondrial E3 ubiquitin ligase MARCH5 also interacts with HBx, targeting it for degradation within the mitochondria. Elevated levels of MARCH5 are associated with improved survival rates in patients with hepatocellular carcinoma (HCC), as its interaction with HBx reduces the oncogenic potential of HBx [80]. Conversely, deubiquitinase JMJD2D helps maintain HBx stability by inhibiting its ubiquitination and proteasomal degradation, thus preserving HBx’s activity in promoting HBV replication and transcription [81].

NAD(P)H: quinone oxidoreductase 1 (NQO1) is significant for stabilizing HBx by preventing its degradation via the 20S proteasome. Inhibition of NQO1 by dicoumarol, which competes with NAD(P)H for binding to NQO1, promotes HBx degradation and restricts cccDNA transcription [82]. Dicoumarol’s ability to degrade HBx highlights its potential as a therapeutic tool for suppressing HBV replication. Nitazoxanide (NTZ), a broad-spectrum anti-infective drug traditionally used to treat protozoal and bacterial infections, also shows promise against HBV. NTZ disrupts the interaction between HBx and DDB1, effectively restoring SMC5 protein levels, which suppresses cccDNA transcription and reduces HBV protein levels [83]. Estradiol benzoate is another novel anti-HBx agent, which is shown to suppress HBV markers. It is believed that estradiol benzoate binds to HBx, potentially preventing the formation of the HBx-DDB1 complex, though further research is needed to fully validate this mechanism [84]. In a screening of 1018 compounds targeting HBx, tranilast was identified as having the most binding affinity for HBx in HBV genotypes B and D. Experimental results demonstrated that tranilast effectively reduced HBV DNA and HBsAg levels without significant cytotoxicity, making it a promising candidate for anti-HBV therapy [85]. Neddylation is indispensable for the enzyme activity of the cullin-RING ligase (CRL) family [86,87,88]. HBX-DDB1-CUL4-ROC1 (CRL4) E3 ligase requires an extra ubiquitin-like protein, NEDD8, for activation [89]. MLN4924 (pevonedistat), an inhibitor of NEDD8-activating enzymes, disrupts HBx-mediated degradation of SMC5/6, significantly inhibiting cccDNA transcription [89,90]. In addition, MLN4924 activates the extracellular signal-regulated kinases (ERK) pathway, which leads to a reduction in key transcription factors required for cccDNA transcription, such as HNF1α, C/EBPα, and HNF4α, further inhibiting HBV antigen expression [91].

## 5. Perspectives

cccDNA is essential for HBV to maintain a persistent infection and represents the biggest barrier to achieving a complete cure for chronic hepatitis B (CHB). As both an intermediate for HBV replication and a template for transcription, cccDNA forms a minichromosome in the nucleus of infected hepatocytes, where it is regulated by host epigenetic and transcriptional factors. Unfortunately, current antiviral therapies for CHB cannot completely eliminate cccDNA, making the development of novel strategies to target cccDNA critical for future HBV cures.

Therapeutic approaches targeting cccDNA include inhibiting its synthesis, employing gene-editing technologies, achieving complete epigenetic silencing, and stimulating the innate immune response in hepatocytes to promote cccDNA degradation. Among these strategies, inhibitors of cccDNA synthesis, such as disubstituted sulfonamides (DSS) CCC-0975 and CCC-0346 [34], ATM/ATR inhibitors [36], and CKII inhibitors [35], have shown promise in significantly limiting the conversion of rcDNA into cccDNA. However, these inhibitors are limited in that they only block the replenishment of the nuclear cccDNA pool and do not affect existing cccDNA. As a result, any remaining cccDNA could still transcribe viral RNA and produce HBV antigens once treatment is discontinued. Epigenetic silencing of cccDNA is another approach, but the numerous epigenetic factors regulating cccDNA chromatin present a challenge. Epigenetic factors associated with cccDNA transcription are not unique, making it unlikely that targeting any single factor will fully inhibit cccDNA transcription. A more effective target may be the HBx protein, which is essential for initiating and sustaining cccDNA transcription. Studies have shown that HBx-deficient HBV lacks transcriptional activity [61,92,93,94]. HBx also upregulates host epigenetic modification enzymes and transcription factors favorable to cccDNA transcription while downregulating host limiting factors through the HBx-DDB1 axis or other mechanisms. Therefore, silencing HBx could potentially achieve complete suppression of cccDNA transcription. Although some HBx inhibitors have shown promise, their safety and efficacy need to be validated by further clinical data. Researchers have also explored the potential of HBx-targeted siRNA therapies for CHB. Experimental data demonstrate significant reductions in HBV markers [95,96,97,98], and siRNA therapy targeting the X region overlapping templates of the HBV genome has shown a favorable hepatic safety profile and efficacy [99]. However, RNAi therapies still require more clinical trials to confirm their safety and long-term effects.

To apply CRISPR-Cas9 in the clinical treatment of chronic hepatitis B (CHB), several critical challenges must be addressed. One of the major concerns is that HBV DNA may integrate into the host genome, posing a risk of chromosomal translocations when using CRISPR-Cas9 to target cccDNA. This potential for unintended genome alterations must be mitigated to ensure patient safety. A more promising approach might be to use base editing to introduce nonsense mutations into cccDNA. By introducing mutations in key regions of the HBV genome, it is likely to completely silence cccDNA, effectively blocking viral transcription and replication. However, a specific liver-targeted delivery system is necessary, and the currently developed lipid nanoparticles (LLNs) and NIR-responsive biomimetic nanoparticles (UCNPs-Cas9@CM) are expected to solve this problem [51,52]. CRISPR activation (CRISPRa) for upregulating APOBEC3A/B and ISG20 is a strategy worth considering to degrade existing cccDNA. However, this approach requires caution, as the APOBEC3 family has been associated with cancer development in some studies [100,101,102,103]. Therefore, the safety of this strategy must be rigorously evaluated in preclinical and clinical settings. Integrated HBV DNA is also a major obstacle to an HBV cure, which is of great significance for the persistent replication of HBV in vivo. A study showed that nucleos(t)ide analogue treatment reduced integrated HBV DNA in twenty-eight patients, but it could not completely eradicate it. Integrated DNA remained in the patients’ bodies even after ten years of treatment [104]. Another trial reported that tenofovir disoproxil fumarate (TDF) treatment markedly decreased integrated and non-integrated HBV DNA of HBV core+ hepatocytes, but had no effect on HBsAg+ hepatocytes [105]. Li et al. completely excised integrated HBV DNA fragments and disrupted the HBV cccDNA in a stable HBV cell line through CRISPR-Cas9, but the results were only visible at the cellular level [106]. In contrast, CRISPR-Cas9-mediated base editors may be a better way to target integrated DNA. The experimental results in the studies of Yang et al. and Smekalova et al. showed that base editors were effective in editing integrated HBV DNA [48,49].

Among the small-molecule drugs that inhibit or degrade cccDNA mentioned in this review, only a small portion of them has entered the clinical trial stage, and the efficacy of most anti-HBV drugs has not been confirmed in humans. For example, in a phase II clinical trial, three different Nitazoxanide (NTZ) treatment regimens with Tenofovir Disoproxil Fumarate (TDF), Tenofovir Alafenamide (TAF), or Entecavir (ETV) all showed a significant reduction in HBsAg levels in CHB patients compared to the placebo group. (https://clinicaltrials.gov/. ID: NCT03905655). In addition, several siRNA drugs targeting HBV transcripts are already in clinical trials, including AB-729, ABI-H0731, RO7445482, and DCR-HBVS. However, for some reason, some trials were terminated without valid data being submitted. There are some promising drugs for HBV, such as ccc_R08 and MLN4924, which do not have clinical trial data, and reliable clinical data are needed to prove the efficacy and safety of these drugs against HBV.

So far, significant achievements have already been made in the treatment of CHB, and many anti-HBV drugs are currently undergoing clinical trials. However, to achieve a complete cure for CHB, it will be essential to develop safe and effective therapies that specifically target or silence cccDNA. In the future, major breakthroughs are bound to be achieved in this field.

## Figures and Tables

**Figure 1 pathogens-13-01100-f001:**
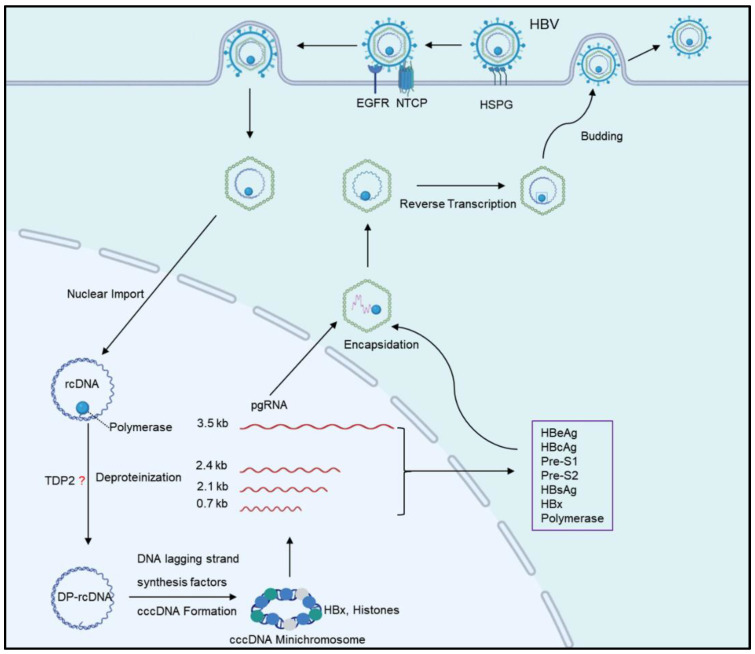
HBV lifecycle. HBV first binds to the low-affinity receptor HSPG, then interacts with NTCP, and enters the liver through endocytosis with the support of EGFR. After removing the nucleocapsid, rcDNA enters the nucleus through the nuclear pore and is subsequently stripped of the polymerase that binds to it, forming DP-rCDNA with the assistance of TDP2. Then, DP-rcDNA utilizes host factors to repair double-stranded gaps and form cccDNA. cccDNA forms minichromosomes with histones and other proteins and is transcribed into HBV transcripts, including pgRNA, with the help of HBx and epigenetic factors. These transcripts are then translated into several viral proteins. pgRNA forms rcDNA through reverse transcription and assembles into virions with viral proteins. Virions are then expelled through exocytosis.

**Figure 2 pathogens-13-01100-f002:**
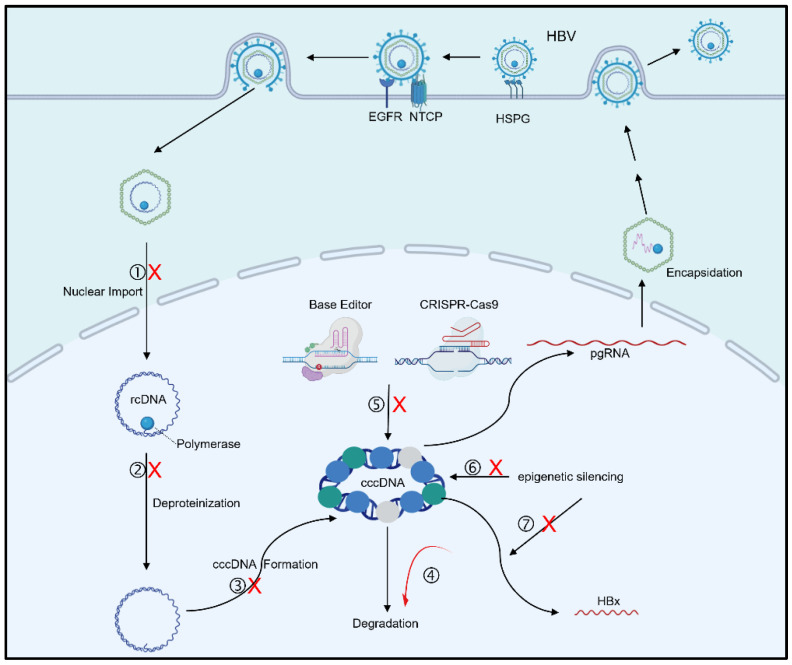
Approaches targeting HBV cccDNA levels. CKII inhibitor DMAT prevents the nuclear import of the polymerase–rcDNA complex. Disubstituted sulfonamides (DSS), CCC-0975 and CCC-0346, disrupt the production of DP-rcDNA and then inhibit cccDNA formation. ATR inhibitors, DNA ligases inhibitors, DNA polymerase inhibitor aphidicolin, FEN1 inhibitor PTPD, and PARP inhibitor Olaparib block the conversion of rcDNA to cccDNA. APOBEC3A/B, UNG2, ISG20, and APEX1 may collaborate to degrade cccDNA. cccDNA reducers, like ccc_R08, decrease cccDNA levels, though their precise mechanisms remain unknown. Although CRISPR-Cas9 KO targeting cccDNA enhances the clearance of cccDNA, it has the potential for off-target effects and poor delivery efficiency. A CRISPR/Cas9-mediated non-cleaving base editor that introduces nonsense mutations into cccDNA is a more promising approach. Epigenetic silencing of cccDNA is another approach, but the numerous epigenetic factors regulating cccDNA minichromosomes present a challenge. A more effective target may be the HBx protein, siRNA, or small-molecule compounds targeting HBx could effectively inhibit cccDNA transcription. ➀ CKII inhibitor DMAT. ➁ Disubstituted sulfonamides (DSS). ➂ ATR inhibitors, DNA ligases inhibitors, DNA polymerase inhibitor, FEN1 inhibitor, PARP inhibitor. ➃ APOBEC3A/B, UNG2, ISG20, APEX1, cccDNA reducers. ➄ CRISPR-Cas9 KO targeting cccDNA, CRISPR/Cas9-mediated non-cleaving base editor. ➅ SETDB1, PRMT5, PRMT1, SIRT3, HMGA1, SMC5/6. ➆ siRNA targeting HBx, Dicoumarol, Nitazoxanide, Estradiol Benzoate, MLN4924 (pevonedistat). The red cross indicates inhibition and the red arrow indicates promotion.

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
