# Peer review of "Targeting HBV cccDNA Levels: Key to Achieving Complete Cure of Chronic Hepatitis B"

_pathogens, 2024, doi:10.3390/pathogens13121100_

Round 1

Reviewer 1 Report

Comments and Suggestions for Authors

This is an extremely detailed and erudite review of molecular mechanisms associated with cccDNA generation and function, along with an extensive survey of the literature regarding possible ways of inhibiting and/or degrading this key barrier to the complete cure of chronic HBV infection. As a clinical virologist with only a passing knowledge of the deep molecular mechanisms underlying cccDNA persistence and transcription, I learnt a considerable amount from reading this manuscript.

My comments are very minor and relate not to the substance of the review, but to possibly improving the presentation. Inevitably, the authors have used a huge number of acronyms and abbreviations relating to the plethora of molecules described. Many of these are correctly spelt out in full with the acronym in brackets when first referred to, but many are not. It is traditionally expected that all abbreviations should be explained at their first usage, but I can understand why this might be difficult in this instance. I have listed below many (but possibly not all) of the abbreviations used (and the line on which they occur) without a full explanation - for some of them at least, I think it would be helpful to explain what they stand for. 

I liked the 2 figures - but suggest that somewhere in the figures or the legend, the authors might indicate that the blue dot shown within the HBV particle is the terminal protein attached to HBV DNA (at least I think that's what it is).

The use of English isn't perfect, but the small number of grammatical errors do not in any way impede understanding of the text, so I would not suggest any changes there.

Finally - the authors mention a large number of agents/molecules/drugs which suppress or degrade cccDNA or other viral markers. I assume the data demonstrating such effects are derived from in vitro studies. I wonder if any of the agents mentioned have undergone any sort of trial in humans yet? A sentence somewhere in the discussion, either listing those drugs with some in-human data or reporting that none of the agents mentioned have yet been used in humans, would be of great interest. Nitazoxanide, for instance, has been used clinically for a number of viral infections - but has it been tried specifically in the treatment of chronic HBV infection?

List of abbreviations without full explanation:

PTPD line 94; PCNA line 104; ATM/ATR line 116; APEX line 134; TREX1 line 135; sgRNA line 186 (gRNA is explained as guide RNA but what does the s stand for?); PHH line 204 (presumably primary human hepatocytes?); IFN-I line 213 - not sure if this is interferon "one" or interferon "letter i"; DDB-1 line 239 - this is explained in full on line 293 but this should be at the first usage; SETDIA line 257; FOXO3alpha, PGC1alpha line 273.

Author Response

Comment 1: [Inevitably, the authors have used a huge number of acronyms and abbreviations relating to the plethora of molecules described. Many of these are correctly spelt out in full with the acronym in brackets when first referred to, but many are not. It is traditionally expected that all abbreviations should be explained at their first usage, but I can understand why this might be difficult in this instance. I have listed below many (but possibly not all) of the abbreviations used (and the line on which they occur) without a full explanation - for some of them at least, I think it would be helpful to explain what they stand for.]

Response: We are very sorry for our carelessness and thank you for pointing this out. According to your valuable comments, we have explained the unclear abbreviations where they first appear. We've highlighted the corrections in our manuscript. (lines 93-94; lines 104-105; line 116; lines 136-138; line 190; lines 214-215; lines 218-219; line 239; lines 245-246; lines 263-264; lines 280-281)

Comment 2: [I liked the 2 figures - but suggest that somewhere in the figures or the legend, the authors might indicate that the blue dot shown within the HBV particle is the terminal protein attached to HBV DNA (at least I think that's what it is).]

Response: Thank you for your valuable suggestions. The HBV DNA polymerase attached to the 5' end of the antisense strand appears in figures but not clearly marked. According to your suggestions, we have added annotations for polymerase in both Figure 1 and Figure 2. (lines 764-842)

Comment 3: [The authors mention a large number of agents/molecules/drugs which suppress or degrade cccDNA or other viral markers. I assume the data demonstrating such effects are derived from in vitro studies. I wonder if any of the agents mentioned have undergone any sort of trial in humans yet? A sentence somewhere in the discussion, either listing those drugs with some in-human data or reporting that none of the agents mentioned have yet been used in humans, would be of great interest. Nitazoxanide, for instance, has been used clinically for a number of viral infections - but has it been tried specifically in the treatment of chronic HBV infection?]

Response: We feel great thanks to your professional comments. Among the small-molecule drugs that inhibit or degrade cccDNA mentioned in this review, only a small portion of them has entered the clinical trial stage, and the efficacy of most anti-HBV drugs has not been confirmed in humans. For example, in a phase II clinical trial, three different Nitazoxanide (NTZ) treatment regimens with Tenofovir Disoproxil Fumarate (TDF), Tenofovir Alafenamide (TAF), or Entecavir (ETV) all showed a significant reduction in HBsAg levels in CHB patients compared to the placebo group. (https://clinicaltrials.gov/. ID: NCT03905655). In addition, several siRNA drugs targeting HBV transcripts are already in clinical trials, including AB-729, ABI-H0731, RO7445482 and DCR-HBVS. However, for some reason, some trials were terminated without valid data being submitted. There are some promising drugs for HBV, such as ccc_R08 and MLN4924, which do not have clinical trial data, and reliable clinical data are needed to prove the efficacy and safety of these drugs against HBV. We've highlighted the corrections and added them in discussion. (lines 401-412)

Reviewer 2 Report

Comments and Suggestions for Authors

This review is well written , I suggest some addenda to improve its content:

1)        How CRISP-Cas9 technology could be applied in therapeutic management of HBV

2)        Do therapeutic approaches against cccDNA  show efficacy against  integrated DNA?

3)        Is there any trails on drugs that have cccDNA  target?

Author Response

Comment 1: [How CRISPR-Cas9 technology could be applied in therapeutic management of HBV]

Response: We sincerely appreciate the valuable comments. Due to the potential integration of HBV DNA into the human genome, the use of traditional CRISPR-Cas9 to eliminate cccDNA is very risky. A more promising approach might be to use base editing to introduce nonsense mutations into cccDNA. However, a specific liver-targeted delivery system is necessary, and the currently developed lipid nanoparticles (LLNs) and NIR-responsive biomimetic nanoparticles (UCNPs-Cas9@CM) are expected to solve this problem. We have discussed this issue in lines 375-384.

Comment 2: [Do therapeutic approaches against cccDNA show efficacy against integrated DNA?]

Response: Considering your great suggestion, we have made supplements in discussion. Integrated HBV DNA is also a major obstacle to HBV cure, which is of great significance for the persistent replication of HBV in vivo. A study showed that nucleos(t)ide analogue treatment reduced integrated HBV DNA in twenty-eight patients, but it could not completely eradicate it. Integrated DNA remained in the patients' body even after ten years of treatment (PMID: 35594553). Another trial reported that tenofovir disoproxil fumarate (TDF) treatment markedly decreased integrated and non-integrated HBV DNA of HBV core+ hepatocytes, but had no effect on HBsAg+ hepatocytes (PMID: 39384203). Li et al. completely excised integrated HBV DNA fragments and disrupted the HBV cccDNA in a stable HBV cell line through CRISPR-Cas9, but the results were only visible at the cellular level (PMID: 28382278). In contrast, CRISPR-Cas9-mediated base editors may be a better way to target integrated DNA. The experimental results in the studies of Yang et al. (PMID: 32278307) and Smekalova et al. (PMID: 32278307) showed that base editors were effective in editing integrated HBV DNA. (lines 389-400)

Comment 3: [Is there any trails on drugs that have cccDNA target?]

Response: We feel great thanks to your professional comments. Among the small-molecule drugs that inhibit or degrade cccDNA mentioned in this review, only a small portion of them has entered the clinical trial stage, and the efficacy of most anti-HBV drugs has not been confirmed in humans. For example, in a phase II clinical trial, Nitazoxanide (NTZ) treatment with Tenofovir Disoproxil Fumarate (TDF), Tenofovir Alafenamide (TAF), or Entecavir (ETV) all showed a significant reduction in HBsAg levels in CHB patients compared to the placebo group. Besides, there are some promising drugs for HBV, such as ccc_R08 and MLN4924, which do not have clinical trial data, and reliable clinical data are needed to prove the efficacy and safety of these drugs against HBV. We have discussed this issue in lines 401-412.